# Cross-sectional survey evaluating the psychological impact of the COVID-19 vaccination campaign in patients with cancer: The VACCINATE study

Daniela Tregnago[1], Alice Avancini[1], Lorenzo Belluomini[1], Ilaria Trestini[1], Marco Sposito[1], Jessica Insolda[1], Federica Bianchi[1], Teodoro Sava[2], Chiara Gaiani[2], Lidia Del Piccolo[3], Valentina Guarnieri[4], Giuseppe Verlato[5], Ahmad Tfaily[5], Roberta Vesentini[5], Serena Zuliani[1], Sara Pilotto[1‡], Michele Milella[1‡]*

1 Section of Innovation Biomedicine - Oncology Area, Department of Engineering for Innovation Medicine (DIMI), University of Verona and University and Hospital Trust (AOUI) of Verona, Verona, VR, Italy, 2 Medical Oncology, Camposampiero Hospital, Padova, Italy, 3 Department of Neuroscience, Psychological and Psychiatric Sciences and Movement Sciences, University of Verona and Verona University Hospital Trust, Verona, Italy, 4 Medical Oncology, Istituto Oncologico Veneto IOV-IRCCS, Padova, Italy, 5 Department of Public Health and Community Medicine, Unit of Epidemiology and Medical Statistics, Istituti Biologici II – University of Verona, Verona, VR, Italy

‡ SP and MM share the last co-authorship on this work.
* michele.milella@univr.it

**Data Availability Statement:** All relevant data are within the manuscript and its Supporting information files.

## Abstract

The COVID-19 pandemic has profoundly impacted on cancer patients' psychological well-being and clinical status. We assessed the levels of anxiety, depression, and distress and the attitude towards COVID-19 vaccination in cancer patients, accepting vaccination at the Verona University Hospital and Camposampiero Hospital in the Veneto region. Self-reported questionnaires were administered to patients undergoing COVID-19 vaccination between March and May 2021 (first and second dose). Twenty-seven items were investigated: i) demographics/clinical characteristics; ii) anxiety, depression, and distress (Hospital Anxiety and Depression Scale—HADS—and Distress Thermometer—DT); iii) four specific items regarding awareness about infection risks, interference with anticancer treatments, and vaccine side effects. Sixty-two and 57% of the patients who accepted to be vaccinated responded to the survey in the two participating Hospitals, respectively. Mean age was 63 years (SD: 12 years; range 19–94 years), women were slightly more prevalent (57.6%), most participants were married (70%), and either worker or retired (60%). Borderline and clinical levels of anxiety were recorded in 14% and 10% of respondents; borderline and clinical levels of depression in 14% and 8%; and moderate and severe distress levels in 33% and 9%. Overall, there was high confidence that vaccination would reduce the risk of contracting COVID-19 (70%), which would make patients feel less worried about contracting the infection (60%). Fear that vaccine-related side effects would interfere with anticancer treatment and/or global health status was low (10% and 9% for items 3 and 4, respectively) and significantly associated with baseline levels of anxiety, depression, and distress at multivariate analysis. Results did not differ between the Verona and Camposampiero cohorts.

**Funding:** This work was supported in part by the ORCHESTRA project; the ORCHESTRA project has received funding from the European Union's Horizon 2020 research and innovation program [grant number 101016167]. The views expressed in this publication are the sole responsibility of the authors, and the Commission is not responsible for any use that may be made of the information it contains. The funders had no role in study design, data collection and analysis, decision to publish, or preparation of the manuscript.

**Competing interests:** The authors declare that they have no conflict of interest.

During the COVID-19 vaccination campaign, adult cancer patients demonstrated high levels of confidence towards vaccination; baseline levels of anxiety, depression, and distress were the only significant predictors of reduced confidence.

## Introduction

Availability of vaccines against severe acute respiratory syndrome coronavirus-2 (SARS-CoV-2) represents a turning point in the war against the COVID-19 pandemic. This has been even more crucial for vulnerable populations, particularly for patients with cancer, where the infection has hit stronger, in terms of both severity and mortality [1]. COVID-19 vaccination recommendations were initially released from national and international Oncology scientific societies on theoretical grounds and based on consensus among peers [2]. Subsequent studies have shown that vaccination is safe and effective in protecting cancer patients from hospitalization and death deriving from COVID-19 infection [3–5]. In Italy, population studies conducted in the Friuli-Venezia Giulia region and Reggio Emilia province show that the risk of death of cancer patients who did not undergo anti-SARS-CoV-2 vaccination is 2–3 fold higher than that of their vaccinated counterpart [6], in line with European data from the OnCovid registry study [7].

While the highly favorable risk/benefit profile is nowadays taken for granted, at the time of vaccination campaigns initiation mistrust in efficacy data, concerns about side effects, and lack of sufficient information led to substantial vaccine hesitancy among cancer patients and the general population alike [8–15]. Even among subjects who felt accepting vaccination was somehow inevitable, reluctance and erroneous perceptions could develop, concurring to increase the psychological burden, particularly in cancer patients in whom the already high level of emotional vulnerability was further impacted by the pandemic [16]. In this context, it is crucial that organizational and protective measures adopted to contain the infection do not worsen patients' mental well-being and that related information is conveyed in a way that reassures patients and elicits their convinced and spontaneous adherence [17]. While the reasons for vaccination refusal have been extensively investigated [18–20], the psychological impact of COVID-19 vaccination in cancer patients who, more or less reluctantly, accepted vaccination has been explored to a much lesser extent [21]. The aim of our study was indeed to investigate the levels of anxiety, depression, and distress, on the one hand, and the subjective perception of the protective effects of the vaccination and its potential interference with anticancer treatment and overall health status, on the other, in cancer patients undergoing COVID-19 vaccination in the context of the campaign promoted by the Veneto Oncology Network (Rete Oncologica Veneta—ROV) between March and May 2021 (the VACCINATE study); here we report the results obtained in two independent cohorts of patients with active cancer vaccinated at a larger academic hub (Verona University and Hospital Trust) and at a smaller local spoke (Camposampiero Hospital).

## Materials and methods

### Study design and participants

A cross-sectional study design was utilized. Data were collected during the COVID-19 vaccination campaign promoted by ROV between March and May 2021, which was directed to adult (≥18 years) patients with active cancer, defined as: *i)* patients with a new cancer diagnosis, scheduled to receive any systemic anticancer treatment; *ii)* patients on ongoing systemic

cancer treatment or who had completed systemic treatment within 6 months from the vaccination proposal. The only mandatory inclusion criteria to participate in the VACCINATE study were: 1) accepting the vaccination proposal, and 2) signing the study-specific informed consent form. Immediately after the Pfizer-BioNTech COVID-19 mRNA vaccine BNT162b2 administration (dose 1 and dose 2), patients were asked whether they would be willing to participate in an anonymous survey investigating their level of anxiety, depression, distress, and perceptions about COVID-19 vaccination and received a copy of the questionnaire to be returned after its completion.

Ethics committee approval was obtained (Prot. No. 80222). Declaration of Helsinki, declaration of Oviedo, as well as the Good Clinical Practice, were followed to conduct the study and design the protocol. STrengthening the Reporting of OBservational studies in Epidemiology (STROBE) statement was followed to report findings [22].

## Questionnaires

An anonymous questionnaire containing 27 items, was drawn after a literature review. The questionnaire was divided into three sections. The first investigated patients' demographics and clinical characteristics, in particular: birth date (open-ended question), sex (male/ female), education level (elementary/up to age 10–11 years; secondary/up to 14 years; secondary/up to 18–19 years; college/university), marital status (single, married, widowed, other), occupational status (worker, retired, student, unemployed, other) tumor site (gastro-intestinal, breast, genito-urinary, lung, melanoma, head/neck, haematological, rare tumor, other) and date of diagnosis (open-ended question). The second section was dedicated to the evaluation of the patient's level of anxiety, depression and distress, using the Hospital Anxiety and Depression Scale (HADS) and the Distress Thermometer (DT). HADS is composed of a total of 14 items (7 items regarding anxiety and 7 items regarding depression) with a 4-point ordinal response format and reports how patients felt in the previous week. Scores for each subscale range from 0 to 10, with scores of 8–10 indicating borderline symptoms, while scoring $\geq 10$ denotes the presence of clinically relevant levels of anxiety and depression [23]. DT is an 11-point numerical analogue scale in which the subject quantified her/his distress [24]; as cutoff scores for specific patient populations may vary [25–27], we elected to analyze DT categorically (null: score 0; mild: scores 1–5; moderate: scores 6–7; or severe: scores $\geq 8$), based on the work of Mitchell and coll. [24]. In the third section, four specific items assessing: *i)* the rational perception of vaccination efficacy (item 1: *"Do you think vaccine can reduce risk of COVID-19 infection and/ or complications?"*), *ii)* subjective feelings towards vaccination protective effects (item 2: *"Do you think the vaccine would make you feel less worried about contracting COVID-19?"*), *iii)* the subjective perception of the possible interference of vaccine side effects with anticancer treatment and global health status, respectively (items 3: *"Are you worried that side effects of COVID-19 vaccine could interfere with your anticancer treatment?"* and 4: *"Are you worried that side effects of COVID-19 vaccine could compromise your health?"*), were developed, using a 4-points Likert scale.

## Statistical analysis

In descriptive statistics, general characteristics were summarized as mean and standard deviation (or median and interquartile range, if the distribution was skewed) for quantitative variables and absolute and percent frequencies for categorical variables. The number of patients that refused the vaccination was calculated. Statistical analysis then considered:

- Primary endpoints: assessment of cancer patients' perception of possible interference of vaccination's side effects with their anti-cancer treatment and global health status (represented by items 3 and 4, respectively);

- Secondary endpoint: assessment of patients' rational perception of vaccination efficacy in reducing infection risks and subjective feelings towards vaccination protective effects (represented by items 1 and 2, respectively).

Significance of the associations between primary or secondary endpoints and the levels of anxiety, depression, and distress, was assessed by chi-squared test or Fisher's exact test. Multivariable analysis was accomplished by ordered logistic regression, where the response to the questions about vaccination was the response variable, while the level of anxiety, depression, and distress were the main variables, and gender, age class, education level, type of cancer, and treatment duration were the potential confounders. Since VACCINATE was designed as a repeated cross-sectional study, first and second-dose patients were different individuals; therefore, socio-demographic and clinical characteristics, as well as basal levels of anxiety, depression, and distress levels, were assessed separately and compared between the two populations (Tables 1 and 2). Since anxiety, depression, and distress levels, as well as answers to the 4 specific items assessing attitudes towards vaccination were homogeneously distributed between the first- and second-dose cohorts, they were considered globally as a single sample for multivariable analysis (Fig 3A, 3B).

## Results and discussion

### Patient population

A total of 1,794 patients with cancer were invited to receive COVID-19 vaccination at the Oncology Unit of Verona University and Hospital Trust, 31 of whom (1.7%) declined the vaccination proposal. One thousand and eighty-nine patients (62%) participated in the anonymous survey investigating their level of anxiety, depression, and distress, as well as perceptions about COVID-19 vaccination, and returned a completed questionnaire after vaccine administration (764 and 325 patients at the first and second dose, respectively). Patients' characteristics are shown in Table 1; no significant differences in socio-demographic and clinical characteristics were observed between patients enrolled at the first and second dose, respectively, with the exception of a higher prevalence of retired patients at the second dose and a different distribution of cancer diagnoses between the first and second dose administrations. Overall, gastrointestinal (GI, 34.7%), breast (24.6%), and genitourinary (GU, 17.3%) cancers accounted for the majority (77%) of cancer diagnoses among responding patients (Table 1).

### Baseline levels of anxiety, depression, and distress

Baseline psychological patients' status was measured using the HADS and DT tools. Borderline and clinical levels of anxiety were detected in 14% and 10% of patients, respectively; the corresponding figures for depression were 14% and 8%; moderate and severe distress levels were observed in 33% and 9%, respectively (Table 2). No significant differences in the distribution of anxiety, depression, and distress were observed according to the dose of vaccination (first or second) at which questionnaires were administered.

The distribution of anxiety, depression, and distress levels according to the underlying cancer diagnosis is shown in Fig 1. Interestingly, clinical anxiety was significantly more frequent in breast cancer and rare tumor patients (17.5% and 21.4%, respectively; $p = 0.005$); although clinical depression also tended to be more frequent among patients with rare tumors (14.8%) and head and neck (9.3%) or breast (8.4%) cancers, these differences did not reach statistical

**Table 1. Sociodemographic and clinical characteristics of the study' participants in Verona.**

| Variable | Total (n = 1,089) N (%) | First administration (n = 764) N (%) | Second Administration (n = 325) N (%) | p-value |
|---|---|---|---|---|
| **Age (years)** | | | | |
| <65 years | 527 (47.3) | 367 (46.8) | 160 (48.5) | .597 |
| ≥65 years | 588 (52.7) | 418 (53.2) | 170 (51.5) | |
| **Gender** | | | | |
| Male | 462 (42.4) | 332 (43.5) | 130 (40.0) | .291 |
| Female | 627 (57.6) | 432 (56.5) | 195 (60.0) | |
| **Education** | | | | |
| University or higher | 162 (15.2) | 114 (15.3) | 48 (15.1) | .520 |
| High school | 482 (45.3) | 329 (44.2) | 153 (48.0) | |
| Junior high | 318 (29.9) | 225 (30.2) | 93 (29.2) | |
| Primary school or lower | 102 (9.6) | 77 (10.3) | 25 (7.8) | |
| **Marital status** | | | | |
| Single | 104 (9.7) | 74 (9.8) | 30 (9.5) | .849 |
| Married | 752 (70.0) | 529 (69.7) | 223 (70.5) | |
| Widowed | 128 (11.9) | 94 (12.4) | 34 (10.8) | |
| Other | 91 (8.5) | 62 (8.2) | 29 (9.2) | |
| **Occupational status** | | | | |
| Worker | 329 (31.4) | 230 (31.3) | 99 (31.7) | < .001 |
| Retired | 298 (28.4) | 165 (22.4) | 133 (42.6) | |
| Student | 6 (0.6) | 3 (0.4) | 3 (1) | |
| Unemployed | 26 (2.5) | 23 (3.1) | 3 (1) | |
| Other | 389 (37.1) | 315 (42.8) | 74 (23.7) | |
| **Tumor site** | | | | |
| Gastro-intestinal | 371 (34.7) | 292 (38.5) | 79 (25.5) | < .001 |
| Breast | 263 (24.6) | 177 (23.4) | 86 (27.7) | |
| Genito-urinary | 185 (17.3) | 117 (15.4) | 68 (21.9) | |
| Lung | 93 (8.7) | 60 (7.9) | 33 (10.7) | |
| Melanoma | 66 (6.2) | 43 (5.7) | 23 (7.4) | |
| Head/Neck | 43 (4.0) | 30 (4.0) | 13 (4.2) | |
| Haematological | 6 (0.6) | 3 (0.4) | 3 (1.0) | |
| Rare tumor | 29 (2.7) | 26 (3.4) | 3 (1.0) | |
| Other | 12 (1.1) | 10 (1.3) | 2 (0.7) | |
| **Time from diagnosis** | | | | |
| ≤18 months | 519 (51.1) | 342 (65.9) | 177 (34.1) | .372 |
| >18 months | 499 (48.9) | 331 (66.3) | 168 (33.7) | |

significance (S1 Table). Anxiety and depression levels were highly correlated with each other (rho = 0.5142, p<0.0001) and with distress levels (rho = 0.4800, p<0.0001 for anxiety and distress, and rho = 0.3962, p<0.0001 for depression and distress; S2 and S3 Tables).

## Attitude towards vaccination

Answers to the 4-item questionnaire exploring attitudes and beliefs towards the risk of contracting COVID-19 and the possibility that vaccination could interfere with oncological treatment and/or global health status were distributed as shown in Fig 2. Overall, there was high confidence that vaccination would reduce the risk of contracting COVID-19 (70%), which would make patients feel less worried about contracting the infection (60%); fear that vaccine-

**Table 2. Levels of anxiety, depression and distress according to COVID-19 vaccine administration.**

| Variable | Total (n = 1089) N (%) | First administration (n = 764) N (%) | Second administration (n = 325) N (%) | p-value |
|---|---|---|---|---|
| **HADS-Anxiety** | | | | |
| Normal | 804 (75.1) | 554 (73.9) | 250 (78.1) | .358 |
| Borderline | 155 (14.5) | 114 (15.2) | 41 (12.8) | |
| Clinical | 111 (10.4) | 82 (10.9) | 29 (9.1) | |
| **HADS-Depression** | | | | |
| Normal | 825 (78.2) | 570 (77.0) | 255 (81.0) | .393 |
| Borderline | 148 (14.0) | 109 (14.7) | 39 (12.4) | |
| Clinical | 82 (7.8) | 61 (8.2) | 21 (6.7) | |
| **Distress Thermometer** | | | | |
| Absent | 223 (21.2) | 158 (21.3) | 65 (20.8) | .655 |
| Mild | 383 (36.4) | 277 (37.4) | 106 (34.0) | |
| Moderate | 351 (33.3) | 239 (32.3) | 112 (35.9) | |
| Severe | 96 (9.1) | 67 (9.0) | 29 (9.3) | |

Note: a HADS classified as follows: score 0–7 points, mild; score 8–10 points, borderline; score ≥10 points, clinically relevant; b Distress thermometer classified as follows: score 0 point, absent; score 1–4 points, mild; score 5–7 points, moderate; score ≥8 points, severe.

related side effects would interfere with anticancer treatment and/or global health status was low (2% and 2%, respectively).

## Associations between patients' characteristics, psychological variables, and attitude towards vaccination

Ordered logistic regression revealed statistically significant associations between the levels of anxiety, depression, and distress and responses to items 3 and 4, assessing the fear that vaccine-related side effects would interfere with anticancer treatment and/or global health status (primary endpoints), respectively, which remained independent at multivariate analysis (Fig 3A and 3B and S4 Table). In particular, cancer patients with borderline and clinical levels of anxiety, borderline levels of depression, and mild to severe levels of distress were significantly more likely to be worried (some or a lot afraid) that COVID-19 vaccination could interfere with their anticancer treatment (item 3, Fig 3A) or with their global health status (item 4, Fig 3B); similarly, female patients were significantly more likely to be worried that COVID-19 vaccination could interfere with their global health status (item 4, Fig 3B) and significantly less confident that vaccination would make them feel less worried about contracting COVID-19 (item 2, S4 Table). Conversely, elderly patients were significantly less likely to be worried about interference between vaccination and global health status (item 4, Fig 3B) and more confident about vaccination protective effects (items 1 and 2, S4 Table). No significant differences in the distribution of answers to items assessing confidence in vaccination efficacy (items 1 and 2) or fear that vaccine-related side effects would interfere with anticancer treatment and/or global health status (items 3 and 4) were observed according to other socio-demographic or clinical characteristics at multivariate analysis (S1 Fig, S4 Table).

## Comparison with patients vaccinated at an independent spoke centre

Five hundred and twenty-nine cancer patients were offered vaccination in the context of the VACCINATE program at the Camposampiero Hospital, 28 (5.3%) of whom declined the vaccination proposal (p-value for the comparison with the Verona population <0.001). Of these,

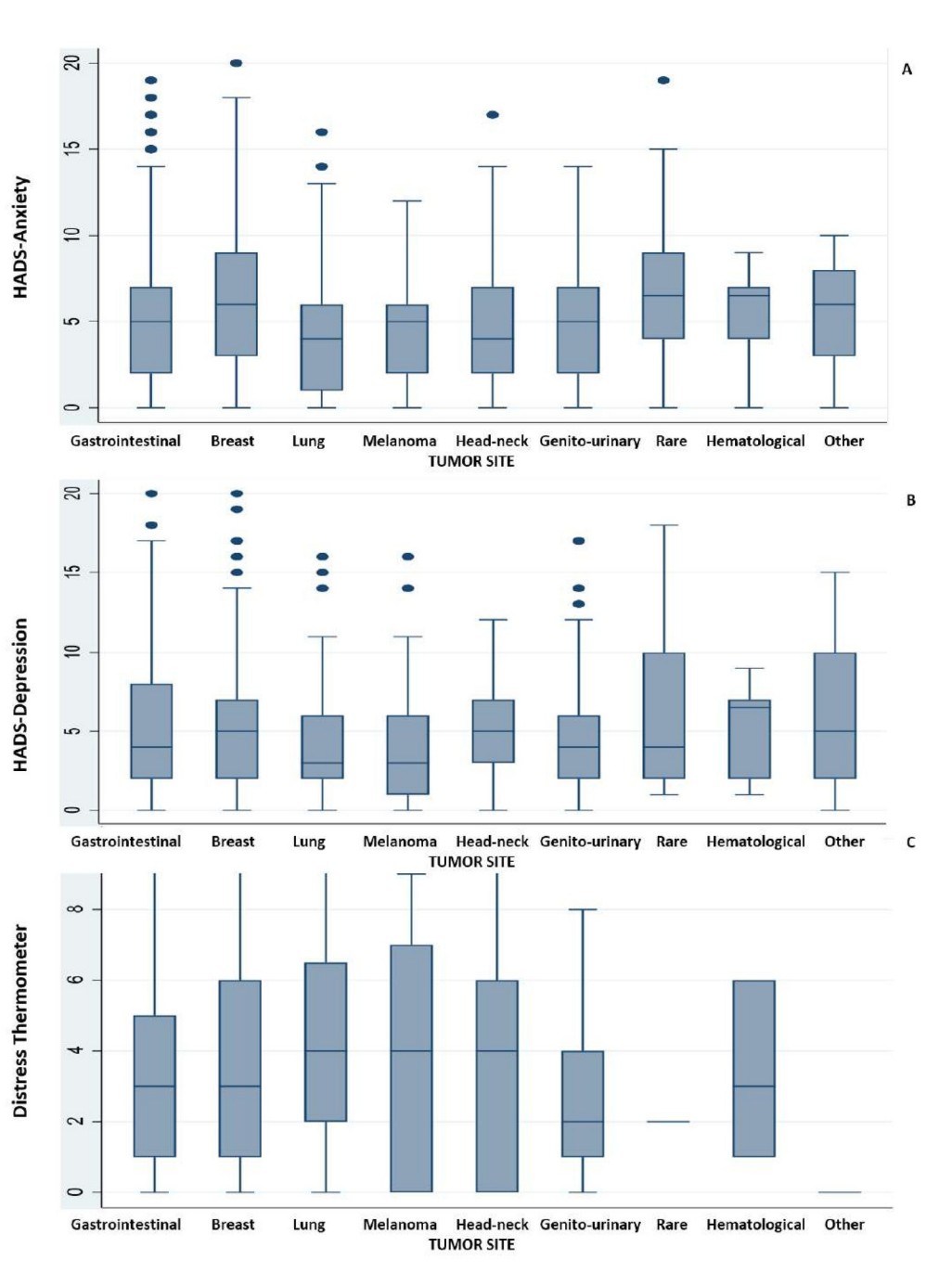

**Fig 1. Distribution of anxiety (A), depression (B), and distress levels (C) according to the underlying cancer sites.**

286 patients (57%) participated in the anonymous survey investigating their levels of anxiety, depression, and distress, as well as perceptions about COVID-19 vaccination, and returned a completed questionnaire (196 and 90 patients after the first and second dose administrations, respectively).

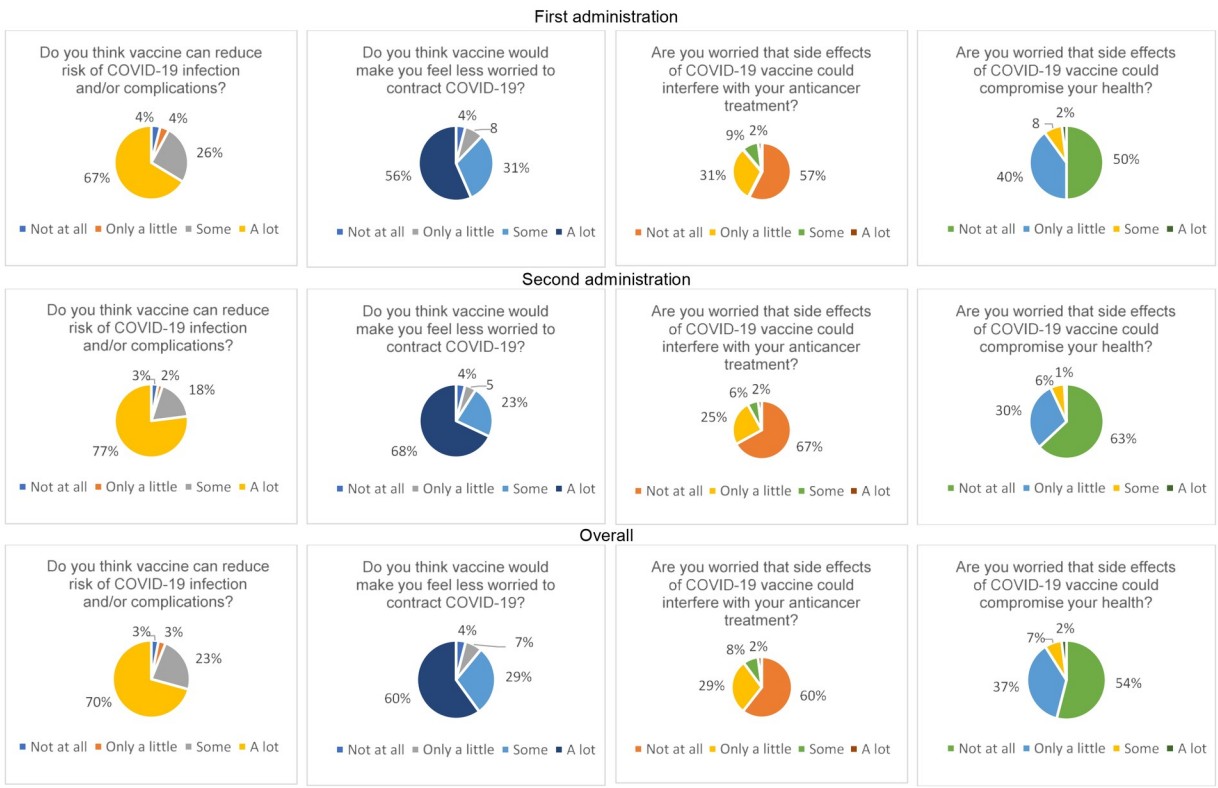

**Fig 2. Answers to the 4-item questionnaire in the first and the second administration.**

Similar to the population of patients assessed in Verona, 14.3% and 5.9% of patients had borderline and clinical levels of anxiety, respectively, and 19.5% and 8.6% had borderline and clinical levels of depression, respectively (S5 Table). Moderate and severe levels of distress were recorded in 30.2% and 9.4% of the Camposampiero population overall, but, at a difference with the Verona population, such distribution significantly shifted towards lower levels of distress among patients who were interviewed at the time of the second dose (p for the comparison between first and second dose in the Camposampiero population = 0.002; S5 Table).

Answers to the 4-item questionnaire were distributed as shown in S2 Fig. Overall, there was high confidence that vaccination would reduce the risk of contracting COVID-19 (52%), which would make patients feel less worried about contracting the infection (48%); fear that vaccine-related side effects would interfere with anticancer treatment and/or global health status was low (5% and 3%, respectively).

In this population of patients, the distribution of responses to item 3 was significantly associated with anxiety levels, with only 13.9% of patients with normal anxiety levels stating that they were some or a lot afraid that *"side effects of COVID-19 vaccine could interfere with [their] anticancer treatment"*, as opposed to 30.6% and 31.5% among patients with borderline and clinical levels of anxiety, respectively (item 3, p = 0.005; S6 Table). Responses to item 4, on the other hand, were significantly influenced by depression levels, with only 8.3% of patients with normal depression levels stating they were some or a lot afraid that *"side effects of COVID-19 vaccine could compromise [their] health"*, as opposed to 26.1% and 27.3% among patients with borderline and clinical levels of depression, respectively (p<0.001; S7 Table). Patients with borderline and clinical depression levels were also significantly less confident that the *"vaccine [could] reduce risk of COVID-19 infection and/or complications"* (item 1, p<0.001; S7 Table).

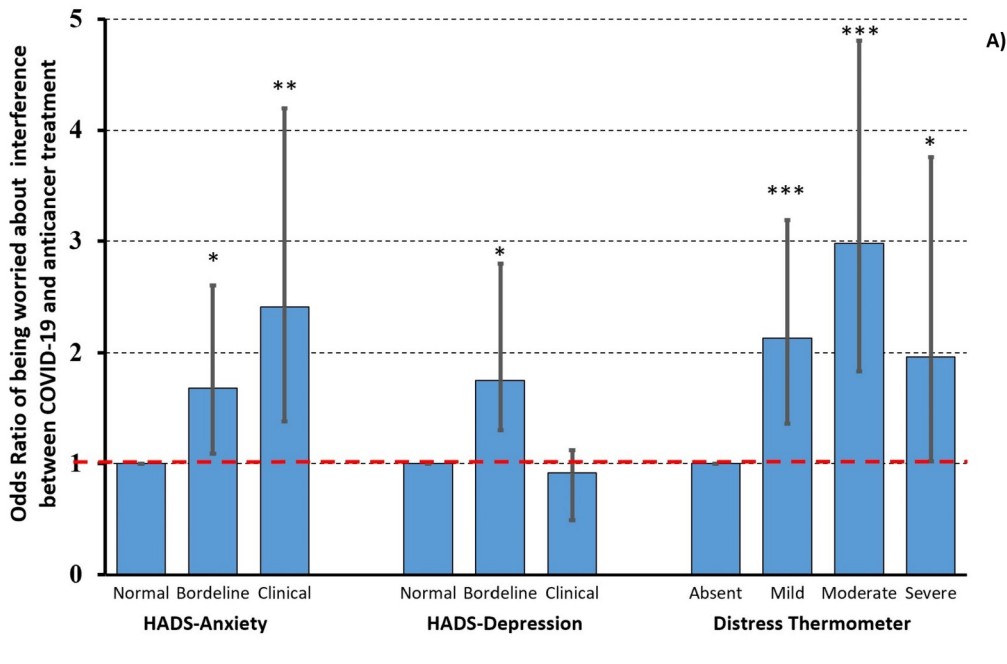

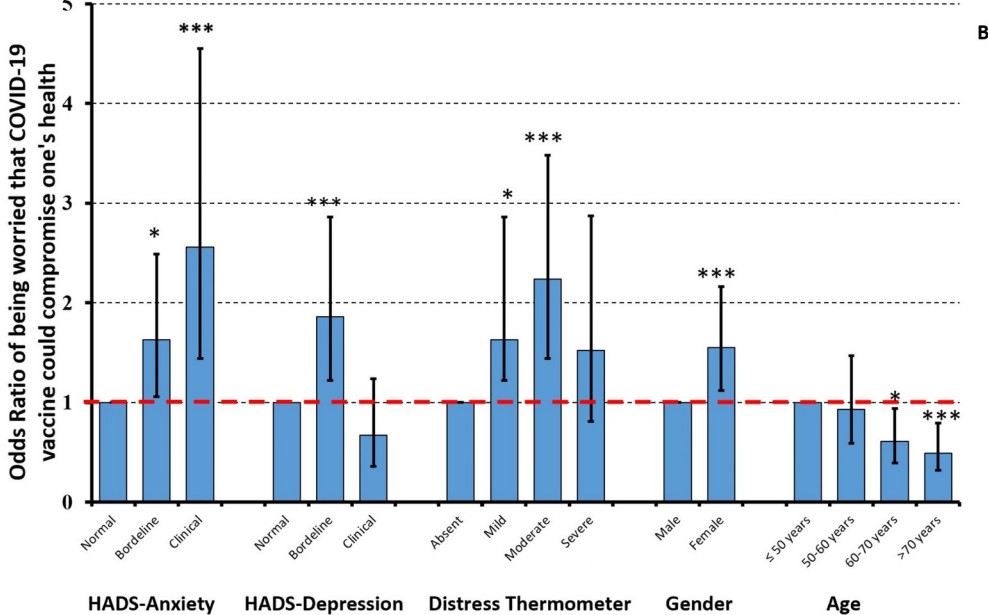

**Fig 3. Influence of potential determinants on being worried that COVID-19 side effects could interfere with anticancer treatments (item3; panel A) or compromise one's health (item 4; panel B).** Odds Ratios (ORs) and 95% confidence intervals were derived by a multivariable logistic regression model. In the graph columns are ORs, bars are 95% confidence intervals. * p = 0.01 ** p = 0.001 *** p<0.001.

## Discussion

In the VACCINATE study, we attempted at correlating the levels of anxiety, depression, and distress with the subjective perception of the protective effects of COVID-19 vaccination and its potential interference with anticancer treatment and overall health status in patients with

active cancer undergoing the first and second vaccine dose. Overall, confidence that vaccination would reduce the risk of contracting COVID-19 was high (70%), resulting in patients feeling less worried about contracting the infection (60%); fear that vaccine-related side effects would interfere with anticancer treatment and/or global health status was low (10% and 9% for items 3 and 4, respectively) and significantly associated with baseline levels of anxiety, depression, and distress at multivariate analysis. Results did not differ in two independent cohorts of patients vaccinated at a larger academic hub (Verona University and Hospital Trust) and at a smaller local spoke (Camposampiero Hospital).

While many studies have investigated the reasons for vaccine refusal and hesitancy in cancer patients [28–33], much less is known about the reciprocal relationships between levels of anxiety, depression, and distress and the attitude towards vaccination in cancer patients who, more or less reluctantly, accept being vaccinated. In our study patients who underwent COVID-19 vaccination, the prevalence of borderline/clinically relevant levels of anxiety (20–24%), depression (22–28%), and moderate/severe distress (39–42%) was consistent with that reported in cancer patients outside a pandemic/vaccination campaign context. A meta-analysis, including 24 studies for a total of 4,007 patients, found that depression affected approximately 15%-16% of patients with oncological diseases, whereas anxiety was present in 10.4% of them [34]; another cross-sectional study shows a slightly higher prevalence (23.4% for depression and 19.9% for anxiety) in a sample of 1,001 patients with mixed cancer types [35]; regarding distress, one patient out of two reported high levels, with fatigue and sleep problems being among the most prevalent associated symptoms [36]. The association of higher clinical anxiety levels with a breast cancer diagnosis observed in our population is also widely confirmed in the literature [36]. With regard to rare neoplasms, on the other hand, there is little supporting data in the literature; however, it is reasonable to speculate that the rarity of the disease itself, which makes prognosis more difficult to evaluate, treatment choices more complicated, and outcome less predictable, would contribute to the state of uncertainty experienced by the patient, thereby triggering clinical anxiety levels.

As reported in recent similar studies [33, 37], the overall confidence in COVID-19 vaccination efficacy in the vaccinated population was high and made patients feeling less worried about contracting the infection. As it might be expected, the main concern in patients with active cancer was the fear that vaccination side effects might interfere with cancer treatment and with their general health status [only 46–60% of respondents were "not (worried) at all", items 3–4]. In the cohort from the *spoke* Centre (Camposampiero Hospital), patients were significantly less distressed and worried at the time of the second administration, perhaps based on not having experienced side effects or delays in their cancer treatment with the first vaccine dose. Although such aspect was not formally investigated in this study, we speculate that the choice of having cancer patients carefully counselled and vaccinated at their treating Oncology services, by physicians and nurses who routinely cared for their cancer, may have substantially contributed to the low hesitancy rate and the favorable psychological profile observed in the VACCINATE study. Other studies have indeed shown that medical recommendations can increase the willingness of patients with cancer to be vaccinated [14–17]. Considering that major concerns about COVID-19 vaccination in oncological populations are related to its safety, side effects, limited efficacy, and interference with cancer and treatments [14, 38], dispelling these doubts may not only push patients towards vaccination, but also help them accept it with a positive attitude and a balanced psychological and emotional status. Interestingly, patients' socio-demographic characteristics only marginally affected their attitude towards vaccination in our study. The most important factors influencing patients' fears about possible detrimental effects of vaccination on their cancer trajectory and general health status were indeed their underlying levels of anxiety, depression, and distress.

Studies conducted in non-cancer patients have evaluated the interference of clinical conditions of anxiety and depression on the attitude towards COVID-19 vaccination. A study in Japan reported that impaired mental health conditions such as depression and generalized anxiety are associated with reluctance or indecision towards vaccination against COVID-19 in the general population [39].

The VACCINATE study has some limitations: indeed, exclusion of patients who refused the vaccination, the inability to collect data on patients who accepted vaccination, but did not agree to participate in the study, may have influenced the results; although the cross-sectional design did not allow longitudinal monitoring of the population, the repeated assessment at the time of both the first and second vaccine administration was able to detect changes in attitude towards vaccination over time, particularly at the *spoke* Centre.

## Conclusions

Despite its limitations, our study provides a thorough assessment of cancer patients' psychological status and attitude towards vaccination under exceptional circumstances (such as the COVID-19 pandemic). In our interpretation, the most important finding is that baseline levels of anxiety, depression, and distress are the major determinant of confidence towards COVID-19 vaccination; together with the observation that anxiety, depression, and distress levels are mostly related to the underlying cancer diagnosis, rather than to the pandemic situation, these data confirm a pressing need to timely and effectively manage cancer patients' emotional and psychological distress, in *normal* and *pandemic* circumstances alike. A meta-analysis including seven randomized controlled trials involving 888 patients found that psychological interventions such as mindfulness-based approaches are effective to manage and relieve anxiety and depression [40]. Additionally, psychological interventions proved to ameliorate treatment-side effects, including fatigue and improve overall patients' quality of life and fear of recurrence [41, 42]. Patients with a well-balanced psychological status, in turn, will be more willing to embrace preventive measures and healthcare recommendations, even under exceptional conditions such as those described here, thereby setting in motion a *virtuous circle*, protecting them from irrational fears (e.g. the so-called "*coronaphobia*" [21]) that may lead to avoiding hospital visits and delaying potentially lifesaving treatments [43]. While further highlighting the irreplaceable role of physician-patient communication in favoring a healthy therapeutic relationship, these data may help improving the penetration and acceptance of vaccination and/or other public health campaigns, not just in cancer patients but also in the general population. Further longitudinal research is needed to monitor the longer-term effects of COVID-19 on psychological health in patients with cancer.

## Supporting information

**S1 Fig. Influence of potential determinants on being worried that COVID-19 side effects could interfere with anticancer treatments (A) or compromise one's health (B).** Odds Ratios (ORs) and 95% confidence intervals were derived by a multivariable logistic regression model. In the graph columns are ORs, bars are 95% confidence intervals. *p = 0.01; **p = 0.001; ***p<0.001.
(TIF)

**S2 Fig. Answers to the 4-item questionnaire in the first and the second COVID-19 vaccine administration at the Spoke center (Camposampiero).**
(TIF)

**S1 Table. Anxiety and depression according to cancer site.**
(DOCX)

**S2 Table. Correlation between HADS-anxiety HADS-depression/DT Distress Thermometer.** *Spearman correlation.
(DOCX)

**S3 Table. Correlation between HADS-anxiety/HADS-depression.** *Spearman correlation.
(DOCX)

**S4 Table. Logistic model of associations of characteristics of cancer patients, anxiety, depression and distress with awareness about COVID-19 infection risks, interference with anticancer treatments, and vaccine side effect.**
(DOCX)

**S5 Table. Psychological status of the study participants at the spoke center (Camposampiero).**
(DOCX)

**S6 Table. Correlation between anxiety levels and responses to the 4-item questionnaire on COVID-19 and vaccination at the spoke center (Camposampiero).**
(DOCX)

**S7 Table. Correlation between the depression levels and responses to the 4-item questionnaire on COVID-19 and vaccination at the spoke center (Camposampiero).**
(DOCX)

## Author Contributions

**Conceptualization:** Daniela Tregnago, Serena Zuliani, Sara Pilotto, Michele Milella.

**Data curation:** Daniela Tregnago, Alice Avancini, Lorenzo Belluomini, Federica Bianchi, Teodoro Sava, Chiara Gaiani, Giuseppe Verlato, Ahmad Tfaily, Roberta Vesentini, Sara Pilotto, Michele Milella.

**Formal analysis:** Daniela Tregnago, Federica Bianchi, Giuseppe Verlato, Roberta Vesentini.

**Funding acquisition:** Sara Pilotto.

**Methodology:** Alice Avancini, Federica Bianchi, Giuseppe Verlato, Roberta Vesentini, Sara Pilotto, Michele Milella.

**Project administration:** Ahmad Tfaily, Sara Pilotto.

**Resources:** Ahmad Tfaily, Roberta Vesentini, Sara Pilotto, Michele Milella.

**Software:** Federica Bianchi, Giuseppe Verlato, Ahmad Tfaily, Roberta Vesentini, Michele Milella.

**Supervision:** Alice Avancini, Ilaria Trestini, Chiara Gaiani, Giuseppe Verlato, Ahmad Tfaily, Roberta Vesentini, Serena Zuliani, Sara Pilotto, Michele Milella.

**Validation:** Daniela Tregnago, Alice Avancini, Lorenzo Belluomini, Ilaria Trestini, Marco Sposito, Jessica Insolda, Federica Bianchi, Teodoro Sava, Chiara Gaiani, Lidia Del Piccolo, Valentina Guarnieri, Giuseppe Verlato, Ahmad Tfaily, Roberta Vesentini, Serena Zuliani, Sara Pilotto, Michele Milella.

**Visualization:** Daniela Tregnago, Lorenzo Belluomini, Ilaria Trestini, Marco Sposito, Jessica Insolda, Federica Bianchi, Teodoro Sava, Chiara Gaiani, Lidia Del Piccolo, Valentina Guarnieri, Giuseppe Verlato, Ahmad Tfaily, Roberta Vesentini, Serena Zuliani, Sara Pilotto, Michele Milella.

**Writing – original draft:** Daniela Tregnago, Alice Avancini, Sara Pilotto, Michele Milella.

**Writing – review & editing:** Daniela Tregnago, Alice Avancini, Lorenzo Belluomini, Ilaria Trestini, Marco Sposito, Jessica Insolda, Federica Bianchi, Teodoro Sava, Chiara Gaiani, Lidia Del Piccolo, Valentina Guarnieri, Giuseppe Verlato, Ahmad Tfaily, Roberta Vesentini, Serena Zuliani, Sara Pilotto, Michele Milella.

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
