## [Decision Letter · Decision Letter 0]

10 May 2023

PONE-D-23-06022A prospective, cross-sectional study evaluating the psychological impact of the COVID-19 vaccination campaign in patients with cancer: The VACCINATE studyPLOS ONE

Dear Dr. Milella,

Thank you for submitting your manuscript to PLOS ONE. After careful consideration, we feel that it has merit but does not fully meet PLOS ONE’s publication criteria as it currently stands. Therefore, we invite you to submit a revised version of the manuscript that addresses the points raised during the review process. Please submit your revised manuscript by Jun 24 2023 11:59PM. If you will need more time than this to complete your revisions, please reply to this message or contact the journal office at plosone@plos.org. Please include the following items when submitting your revised manuscript:A rebuttal letter that responds to each point raised by the academic editor and reviewer(s). You should upload this letter as a separate file labeled 'Response to Reviewers'.A marked-up copy of your manuscript that highlights changes made to the original version. You should upload this as a separate file labeled 'Revised Manuscript with Track Changes'.An unmarked version of your revised paper without tracked changes. You should upload this as a separate file labeled 'Manuscript'.

We look forward to receiving your revised manuscript.

Kind regards,

Suzanne Rose, PhD

Academic Editor

PLOS ONE

   "YES. This work was supported by the ORCHESTRA project; the ORCHESTRA project has received funding from the European Union’s Horizon 2020 research and innovation program [grant number 101016167]."

Additional Editor Comments:

Please carefully review for readability in the English language to ensure sentence structure is solid and understandable. Please also avoid abbreviating pts or pt throughout and instead use the word patient or patients. 

I appreciate the novelty of the research you have provided and invite you to review the comments below from the two reviewers. I also do apologize for the length of time spent in securing reviewers for this important work. Thank you for your patience as high-quality reviewers were secured to provide the best possible feedback to your work. 

Reviewers' comments:

Reviewer's Responses to Questions

**Comments to the Author**

1. Is the manuscript technically sound, and do the data support the conclusions?

Reviewer #1: Partly

Reviewer #2: Yes

2. Has the statistical analysis been performed appropriately and rigorously? 

Reviewer #1: I Don't Know

Reviewer #2: Yes

3. Have the authors made all data underlying the findings in their manuscript fully available?

Reviewer #1: Yes

Reviewer #2: Yes

4. Is the manuscript presented in an intelligible fashion and written in standard English?

Reviewer #1: No

Reviewer #2: Yes

5. Review Comments to the Author

Reviewer #1: General comment: Thank you for the opportunity to revise the manuscript entitled “A prospective, cross-sectional study evaluating the psychological impact of the COVID-19 vaccination campaign in patients with cancer: The VACCINATE study”.

This paper surveyed a large cohort of adult cancer patients on their anxiety and distress symptomatology and attitude toward a COVID-19 vaccination. Participants were identified at two oncology wards in Italy (Verona and Veneto region). Patient reported outcomes were measured in patients who underwent the first or second dose of COVID-19 vaccination.

The topic is a valuable one and the paper can provide a contribution to the literature in this area, although there are some limitations in terms of methodology and results.

Comment 1: The article is reasonably well organized and written. While quite well-written in English, throughout the paper there are some linguistic inaccuracies and imprecision in language that makes the meaning unclear (esp. in the methods section). Editing for clarity of communication is needed.

Comment 2: The title is misleading since the term “prospective, cross-sectional study” is

contradictory.

Comment 3: Regarding Abstract/Background: Please name the term (patients) in full first before abbreviating it. The term "psychological status" is somewhat inaccurate and should be replaced with a more precise term that better describes the constructs being measured. Regarding Abstract/Methods and findings: Please add some information on patient sample size, response rate and sociodemographic characteristics like median age and sex.

Comment 4: Regarding Introduction: The authors occasionally speak of "a substantial body of evidence" or "a growing body of evidence" (p.3., ll. 72/92) but refer only to a single citation, which does not really support the claim thus made.

Comment 5: Regarding Introduction: The term "active cancer patient" (p.3, l. 99) is somehow misleading. "Patients with active cancer" does seem to be more appropriate.

Comment 6: Regarding Introduction: The authors should clarify the objective of the study, as it is not sufficiently compelling to state that "results will be reported" without stating the specific intent.

I am not sure what "patient characteristics" means. The term should be described in more detail, i.e., whether it is sociodemographic and clinical characteristics.

Comment 7: Regarding Methods, Study Design and participants: The authors stated that they have assessed patients who refused the vaccination. Were additional variables collected describing them in terms of age, sex, tumor-specific variables to compare them to patients who received the vaccination?

Please move the sentence "the number of patients who refused vaccination was calculated" to the statistical analyses section (p. 4, line 112).

Comment 8: Regarding Methods, Study Design and participants: The authors should specify the eligibility criteria ii) confirmed diagnosis of cancer, since the term “active cancer” was previously introduced. Where there any exclusion criteria in terms of treatment, cancer site, stage, etc.?

Comment 9: Regarding Methods, Questionnaires: Please provide the appropriate references for the questionnaires used.

Is there a rationale for why the metric scale of the DT was structured categorically? Research support a cut-off score of 5 on the DT to indicate patients with clinically elevated distress levels.

I don't quite understand how the item "Do you think the vaccine makes you less afraid of contracting COVID-19?" is supposed to represent an interference with cancer treatments.

Comment 10: Regarding Methods, Statistical analysis: Please change the term “categorical variables” to “continuous variables” (p.5, l.150). Item number 3 mentioned in the statistical analysis for the primary outcome does not match with the item number from the description of study measures (where “interferences with anticancer treatments” refers to item number 2).

It is unclear what the authors meant by “significant interference” (p.5, l.158) – please clarify.

Some expressions used to describe the statistical analyses seem inappropriate, i.e. “main determinants”.

This section contains the objectives of the study, which should be mentioned in the last paragraph of the introduction. This section needs to be revised for a detailed description of all statistical analyses used to test the research questions.

The authors considered two independent samples (first and second vaccination) as one sample (p.5, ll.163-66). It is unclear what the authors mean by this, as the following results are based on comparisons between the two groups.

Comment 11: Regarding results, patient sample: p.5-6, ll. 174-78 This sentence needs rephrasing. It is not clear what the authors mean by ”significant imbalances”. Please clarify.

Comment 12: Regarding results, table 1 and other tables including p values: Please do not include the zero before the decimalpoint reporting the p values.

Comment 13: Regarding results, suppl. Table S3: The abbreviation of the distress thermometer must be DT, not TD. Please rectify. The decimals points of the p value amounts to three not four as displayed in table S3.

Comment 14: Regarding results, Figure 2: It was a little difficult to capture the diagrams. Consider a similar color scheme for similar items to make visual comparisons easier.

Comment 15: Regarding results, logistic regression analysis: The heading does not summarize the content of the section below it. It is not clear how the variables are related to each other (direction of the correlation). To me, it is confusing that response frequencies are reported within the results of the regression analysis. This section needs restructuring.

Comment 16: Regarding results, comparison of hospitals: It would certainly be interesting to describe the patient characteristics of the second sample from Camposampiero Hospital.

Comment 17: Regarding discussion: I’m not sure what the term “overall figures” means. Please clarify.

Please remove the references for tables in the discussion.

Comment 18: Regarding discussion: Unfortunately, a detailed description of the nature and significance of the results and their implications is lacking. The authors should better elaborate the implications for practice and further research needs.

Reviewer #2: I have few comments for the authors:

1) It is not clear the meaning of "Prospective.." in the title given that the study design is cross-sectional. I suggest to delete it from the title.

2) Number of enrolled patients: if I correctly understood, out of 1794 patients invited for vaccination, 1763 accepted vaccination and 1089 accepted to take part in the study-i.e., a participation rate of 61.8%. What are the characteristics of vaccinated patients that refuse study participation? The authors should comment on this potential selection bias.

3) The authors made hundreds of statistical tests: a comment on the role of multiple testing before acceptance of statistical significant differences would be welcomed.

6. PLOS authors have the option to publish the peer review history of their article (what does this mean?). If published, this will include your full peer review and any attached files.

Reviewer #1: No

Reviewer #2: No

---

## [Author Response · Author response to Decision Letter 0]

29 Jul 2023

Verona, July 27th, 2023

Prof. Suzanne Rose, PhD

Dear Editor,

Enclosed please find a thoroughly revised version of the manuscript entitled “Cross-sectional survey evaluating the psychological impact of the COVID-19 vaccination campaign in patients with cancer: The VACCINATE study" [PONE-D-23-06022] that we wish to resubmit for publication in Plos One. 

We would like to thank the reviewers for the thoughtful and stimulating comments that have prompted us to update and clarify several points and revise the manuscript accordingly. 

We thus hope that the quality of the manuscript has now substantially improved, so that it may be reconsidered for publication.

A point-by-point rebuttal description of the performed revisions follows herein:

 Editor

 Editor’ Comments: Response:

1) PLOS ONE's style requirements Done

2) 

State what role the funders took in the study The following comment was added to the financial disclosure statement (p.15, ll. 514/517). The funders had no role in study design, data collection and analysis, decision to publish, or preparation of the manuscript.

3) ORCID iD for the corresponding author The link to automatically link the OrcidID to the submission does not work properly; the OrcidID for the corresponding author (prof. Michele Milella) is: 0000-0002-3826-5237 

4) We note that you have included the phrase “data not shown” in your manuscript. Unfortunately, this does not meet our data sharing requirements. PLOS does not permit references to inaccessible data. Data not shown in the previous version of the manuscript have now been provided as Supplementary Tables S1

5) Include captions for your Supporting Information files at the end of your manuscript, and update any in-text citations to match accordingly 

Done

 Referee #1 

 Reviewer's Comments: Response:

 General comments. We thank the reviewer for acknowledging the potential interest of our piece of work and for giving us important suggestions on how to improve the manuscript.

1) The article is reasonably well organized and written. While quite well-written in English, throughout the paper there are some linguistic inaccuracies and imprecision in language that makes the meaning unclear (esp. in

the methods section). Editing for clarity of communication is needed. 

We thank the reviewer for his/her suggestion. Language editing has been implemented throughout the manuscript. 

2) The title is misleading since the term “prospective, cross-sectional study” is

contradictory. According to the reviewer's suggestion, we have revised the title into: Cross-sectional survey evaluating the psychological impact of the COVID-19 vaccination campaign in patients with cancer: The VACCINATE study 

3) Regarding Abstract/Background: Please name the term (patients) in full first before abbreviating it. The term "psychological status" is somewhat inaccurate and should be replaced with a more precise term that better describes the constructs being measured. Regarding Abstract/Methods and findings: Please add some information on patient sample size, response rate and sociodemographic characteristics like median age and sex. 

We have thoroughly revised the abstract, according to the reviewer's suggestions. 

4) Regarding Introduction: The authors occasionally speak of “a substantial body of evidence” or “a growing body of evidence” (p.3., ll. 72/92) but refer only to a single citation, which does not really support the claim thus made. According to the reviewer's suggestion, we have rephrased the relative paragraphs of the Introduction and added updated relevant references.

5) The term "active cancer patient" (p.3, l. 99) is somehow misleading. "Patients with active cancer" does seem to be more appropriate The term "patients with active cancer" was substituted for "active cancer patients" throughout the manuscript, according to the reviewer's suggestion.

6) The authors should clarify the objective of the study, as it is not sufficiently compelling to state that "results will be reported" without stating the specific intent.

I am not sure what "patient characteristics" means. The term should be described in more detail, i.e., whether it is sociodemographic and clinical characteristics According to the reviewer's suggestion, we have rephrased the paragraph of the Introduction regarding study aims. 

7) The authors stated that they have assessed patients who refused the vaccination. Were additional variables collected describing them in terms of age, sex, tumor-specific variables to compare them to patients who received the vaccination?

Please move the sentence "the number of patients who refused vaccination was calculated" to the statistical analyses section (p. 4, line 112). We thank the reviewer for his/her comment. The study was designed in such a way that accepting the vaccination proposal and signing the informed consent form were the only mandatory inclusion criteria; this, unfortunately, precluded the possibility to collect information on patients refusing vaccination. 

The sentence on the number of patients who refused vaccination has been moved to statistical analyses section as suggested.

Data were collected in the context of the COVID-19 vaccination campaign promoted by ROV between March and May 2021, which was directed to adult (>18 years) patients with active cancer, defined as: patients with a new cancer diagnosis, scheduled to receive any systemic anticancer treatment; patients on ongoing systemic cancer treatment or who had completed systemic treatment within 6 months from the vaccination proposal. The only mandatory inclusion criteria to participate in the VACCINATE study were: 1) accepting the vaccination proposal, and 2) signing the study-specific informed consent form. 

Eligibility criteria for both the vaccination campaign and the VACCINATE study have now been clarified in the Methods section of the revised manuscript (p.4, ll. 107/113).

8) The authors should specify the eligibility criteria ii) confirmed diagnosis of cancer, since the term “active cancer” was previously introduced. Where there any exclusion criteria in terms of treatment, cancer site, stage, etc.? 

9) Regarding Methods, Questionnaires: Please provide the appropriate references for the questionnaires used.

Is there a rationale for why the metric scale of the DT was structured categorically? Research support a cut-off score of 5 on the DT to indicate patients with clinically elevated distress levels.

I don't quite understand how the item "Do you think the vaccine makes you less afraid of contracting COVID-19?" is supposed to represent an interference with cancer treatments. We thank the reviewer for his/her suggestions. 

We have now appropriately referenced the questionnaires utilized. 

Concerning the metric scale of the DT, as cutoff scores for specific patient populations may vary, we elected to analyze DT categorically, based on the work of Mitchell and coll. [ref. 24]. This has now been clearly stated in the Methods section of the revised manuscript (p.4, ll. 132/135).

Item 2 has been developed as complementary to Item 1, in that Item 1 asks whether the subject rationally believes that vaccination could reduce the risk of contracting COVID-19 and Item 2 asks whether this would make him/her less afraid. Both items reflect the subject's confidence that vaccination would avoid contracting COVID-19 and therefore not being able to access the hospital for anticancer treatment. This has been more clearly explained in the Methods section of the revised manuscript (p.4, ll. 137/138).

10) Regarding Methods, Statistical analysis: Please change the term “categorical variables” to “continuous variables” (p.5, l.150). Item number 3 mentioned in the statistical analysis for the primary outcome does not match with the item number from the description of study measures (where “interferences with anticancer treatments” refers to item number 2). It is unclear what the authors meant by “significant interference” (p.5, l.158) – please clarify.

Some expressions used to describe the statistical analyses seem inappropriate, i.e. “main determinants”.

This section contains the objectives of the study, which should be mentioned in the last paragraph of the introduction. This section needs to be revised for a detailed description of all statistical analyses used to test the research questions.

The authors considered two independent samples (first and second vaccination) as one sample (p.5, ll.163-66). It is unclear what the authors mean by this, as the following results are based on comparisons between the two groups. According to the reviewer's suggestions, we have extensively revised Statistical analysis section of the Methods, eliminating imprecisions and correcting statistical terms, where appropriate.

We have also more clearly explained the rationale for treating the first- and second-dose populations as either two separate cohorts or a single sample (p.5, ll. 160/163).

11) Regarding results, patient sample: p.5-6, ll. 174-78 This sentence needs rephrasing. It is not clear what the authors mean by “significant imbalances”. Please clarify. The sentence has been rephrased according to the reviewer's suggestion (p.6, ll. 186/188) of the revised manuscript).

12) Regarding results, table 1 and other tables including p values: Please do not include the zero before the decimal point reporting the p values. We have corrected the tables accordingly, if this is compliant with journal style.

13) Regarding results, suppl. Table S3: The abbreviation of the distress thermometer must be DT, not TD. Please rectify. The decimals points of the p value amounts to three not four as displayed in table S3 We have corrected Supplementary Table S3 according to the reviewer's comments.

14) Regarding results, Figure 2: It was a little difficult to capture the diagrams. Consider a similar color scheme for similar items to make visual comparisons easier We have revised Figure 2 according to the reviewer's comments.

15) Regarding results, logistic regression analysis: The heading does not summarize the content of the section below it. It is not clear how the variables are related to each other (direction of the correlation). To me, it is confusing that response frequencies are reported within the results of the regression analysis. This section needs restructuring. We thank the reviewer for his/her extremely useful comment; following his/her advice, we have completely re-written the paragraph (page 6-7, lines 208-226 of the revised manuscript), provided additional Figures (Figure 3A-B and Supplementary Figure S1A-B of the revised manuscript), moved former Table 3 in the Supplementary materials (Supplementary Table S4 of the revised manuscript), and eliminated several Supplementary tables (former Supplementary Tables S4, S5, S6). We believe these changes will contribute to clarifying the nature and meaning of the data being presented and the overall readability of the manuscript. 

16) Regarding results, comparison of hospitals: It would certainly be interesting to describe the patient characteristics of the second sample from Camposampiero Hospital We completely agree with the reviewer's comment; unfortunately, it was not possible to retrieve the socio-demographic characteristics of patients vaccinated at the Camposampiero Hospital.

17) Regarding discussion: I’m not sure what the term “overall figures” means. Please clarify.

Please remove the references for tables in the discussion. According to the reviewer's suggestion, we have thoroughly revised the Discussion section, highlighting strengths and limitations, our interpretation of the results, implications for practice, and further research needs in the final paragraph (p.9-10, ll. 293/344) of the revised manuscript).

18) Regarding discussion: Unfortunately, a detailed description of the nature and significance of the results and their implications is lacking. The authors should better elaborate the implications for practice and further research needs. 

 Referee #2

 Reviewer's Comments: Response:

1) It is not clear the meaning of "Prospective..." in the title given that the study design is cross-sectional. I suggest to delete it from the title. According to the reviewer's suggestion, we have revised the title into: Cross-sectional survey evaluating the psychological impact of the COVID-19 vaccination campaign in patients with cancer: The VACCINATE study

2) 

Number of enrolled patients: if I correctly understood, out of 1794 patients invited for vaccination, 1763 accepted vaccination and 1089 accepted to take part in the study-i.e., a participation rate of 61.8%. What are the characteristics of vaccinated patients that refuse study participation? The authors should comment on this potential selection bias. We thank the reviewer for his/her comment. The study was designed in such a way that accepting the vaccination proposal and signing the informed consent form were mandatory inclusion criteria; this, unfortunately, precluded the possibility to collect information on patients refusing vaccination or patients accepting vaccination, but not agreeing to participate in the study; such limitation is now clearly acknowledged in the Discussion section (p.9-10, ll. 318/323 of the revised manuscript).

3) The authors made hundreds of statistical tests: a comment on the role of multiple testing before acceptance of statistical significant differences would be welcomed We thank the reviewer for his/her suggestion. Adjusting for multiple testing bias is mandatory in confirmatory analyses, such as randomized controlled trials, but not in exploratory analyses, such as observational exploratory studies. Anyway, to cope with this problem we identified primary endpoints (items 3 and 4), and secondary endpoints (items 1 and 2), and we particularly focused on the former. Moreover, we performed both univariable analyses where single variables are tested one by one, and multivariable analysis, where the most relevant variables are evaluated altoghether by ordered logistic regression models. Hence, the risk of possible multiple testing bias, which could affect univariable analysis, was dealt with in the final multivariable analysis. For all the above-mentioned reasons, we do not think that the use of multiplicity adjustment, such as Šidák’s or Bonferroni’s corrections, would significantly improve the present manuscript, but are willing to implement them, should the Editor deem it appropriate.

Moreover, we have checked carefully the author checklist in order to provide all the requested manuscript details.

We look forward to hearing from you regarding our submission and hope that the manuscript is now acceptable for publication in your prestigious Journal and of interest to PLOS One readers. We would be glad to respond to any further questions and comments that you may have. 

Best regards,

Michele Milella, Sara Pilotto, and Daniela Tregnago on behalf of all authors.

---

## [Decision Letter · Decision Letter 1]

16 Aug 2023

Cross-sectional survey evaluating the psychological impact of the COVID-19 vaccination campaign in patients with cancer: The VACCINATE study

PONE-D-23-06022R1

Dear Dr. Milella,

We’re pleased to inform you that your manuscript has been judged scientifically suitable for publication and will be formally accepted for publication once it meets all outstanding technical requirements.

Kind regards,

Suzanne Rose

Academic Editor

PLOS ONE

Additional Editor Comments (optional): Thank you for adequately addressing all comments and revising the manuscript as requested. 

Reviewers' comments:

Reviewer's Responses to Questions

**Comments to the Author**

1. If the authors have adequately addressed your comments raised in a previous round of review and you feel that this manuscript is now acceptable for publication, you may indicate that here to bypass the “Comments to the Author” section, enter your conflict of interest statement in the “Confidential to Editor” section, and submit your "Accept" recommendation.

Reviewer #2: All comments have been addressed

2. Is the manuscript technically sound, and do the data support the conclusions?

Reviewer #2: Yes

3. Has the statistical analysis been performed appropriately and rigorously? 

Reviewer #2: Yes

4. Have the authors made all data underlying the findings in their manuscript fully available?

Reviewer #2: Yes

5. Is the manuscript presented in an intelligible fashion and written in standard English?

Reviewer #2: Yes

6. Review Comments to the Author

Reviewer #2: The authors have made a sound revision of the article and they have answered to my comments. - I do not have further comments

7. PLOS authors have the option to publish the peer review history of their article (what does this mean?). If published, this will include your full peer review and any attached files.

Reviewer #2: No

---

## [Editor Report · Acceptance letter]

11 Sep 2023

PONE-D-23-06022R1 

Cross-sectional survey evaluating the psychological impact of the COVID-19 vaccination campaign in patients with cancer: The VACCINATE study. 

Dear Dr. Milella:

I'm pleased to inform you that your manuscript has been deemed suitable for publication in PLOS ONE. Congratulations! Your manuscript is now with our production department. 

Kind regards, 

on behalf of

Dr. Suzanne Rose 

Academic Editor

PLOS ONE